# Distribution, antifungal susceptibility pattern and intra-*Candida albicans* species complex prevalence of *Candida africana*: A systematic review and meta-analysis

**Sanaz Aghaei Gharehbolagh[1], Bahareh Fallah[2], Alireza Izadi[1], Zeinab Sadeghi Ardestani[2], Pooneh Malekifar[3], Andrew M. Borman[4], Shahram Mahmoudi[5]***

**1** Department of Medical Parasitology and Mycology, School of Public Health, Tehran University of Medical Sciences, Tehran, Iran, **2** Department of Mycology, Faculty of Medical Sciences, Tarbiat Modares University, Tehran, Iran, **3** Department of Epidemiology and Biostatistics, School of Public Health, Tehran University of Medical Sciences, Tehran, Iran, **4** Public Health England UK National Mycology Reference Laboratory, Southmead Hospital Bristol, Medical Research Council Centre for Medical Mycology (MRC CMM), University of Exeter, Exeter, United Kingdom, **5** Department of Medical Parasitology and Mycology, School of Medicine, Iran University of Medical Sciences, Tehran, Iran

\* Mahmoudi.sh@iums.ac.ir, sh.mahmoudi93@gmail.com

## Abstract

*Candida africana* is a pathogenic species within the *Candida albicans* species complex. Due to the limited knowledge concerning its prevalence and antifungal susceptibility profiles, a comprehensive study is overdue. Accordingly, we performed a search of the electronic databases for literature published in the English language between 1 January 2001 and 21 March 2020. Citations were screened, relevant articles were identified, and data were extracted to determine overall intra-*C. albicans* complex prevalence, geographical distribution, and antifungal susceptibility profiles for *C. africana*. From a total of 366 articles, 41 were eligible for inclusion in this study. Our results showed that *C. africana* has a worldwide distribution. The pooled intra-*C. albicans* complex prevalence of *C. africana* was 1.67% (95% CI 0.98–2.49). Prevalence data were available for 11 countries from 4 continents. Iran (3.02%, 95%CI 1.51–4.92) and Honduras (3.03%, 95% CI 0.83–10.39) had the highest values and Malaysia (0%) had the lowest prevalence. Vaginal specimens were the most common source of *C. africana* (92.81%; 155 out of 167 isolates with available data). However, this species has also been isolated from cases of balanitis, from patients with oral lesions, and from respiratory, urine, and cutaneous samples. Data concerning the susceptibility of *C. africana* to 16 antifungal drugs were available in the literature. Generally, the minimum inhibitory concentrations of antifungal drugs against this species were low.

In conclusion, *C. africana* demonstrates geographical variation in prevalence and high susceptibility to antifungal drugs. However, due to the relative scarcity of existing data concerning this species, further studies will be required to establish more firm conclusions.

**Data Availability Statement:** All relevant data are within the manuscript and its Supporting Information files.

**Funding:** The authors received no specific funding for this work.

**Competing interests:** The authors have declared that no competing interests exist.

## Introduction

The medically important polyphyletic genus *Candida* contains more than 300 different yeast species, around 20 of which are regularly reported from human infections ranging in spectrum from superficial mycoses to deep-seated and disseminated infections [1–3]. *Candida albicans* is widely accepted as the most virulent species in the genus, and is the etiological agent in approximately 50%, 95%, and 80–90% of cases of nosocomial bloodstream *Candida* infections, oropharyngeal and vulvovaginal candidiasis, respectively [4–7].

*C. albicans* is a complex of three closely-related species, *C. albicans sensu stricto*, *C. dubliniensis*, and *C. africana* [6, 8]. *C. africana*, which was first isolated in Africa in 1995, was proposed as a new species within the *C. albicans* complex in 2001 [9, 10]. With a worldwide distribution, *C. africana* has been isolated from diverse clinical specimens (mucous membranes, cutaneous samples, specimens from the urinary and respiratory tracts, blood) and has been reported to cause a wide variety of human infections including vulvovaginal candidiasis, oral thrush, and blood stream infections. [11–15].

Unlike the other members of *C. albicans* complex, *C. africana* is unable to form chlamydospores and cannot assimilate glucosamine, N-acetylglucosamine, trehalose, or DL-lactate. However, in common with *C. albicans* and *C. dubliniensis* it has retained the capacity to produce germ-tubes. Moreover, molecular studies have demonstrated high levels of genetic relatedness between *C. africana* and *C. albicans* [16–18]. Thus, differentiation of *C. africana* from the other members of *C. albicans* complex using conventional identification techniques is difficult [19, 20].

Given these issues, molecular methods such as an end point PCR based on size polymorphism of the *hwp1* gene (*C. albicans*: 941bp, *C. dubliniensis*: 569 bp, and *C. africana*: 700 bp) have been designed to discriminate between *C. albicans*, *C. dubliniensis*, and *C. africana* [21]. Using such approaches, the prevalence of *C. africana* within the *C. albicans* complex has been reported to vary significantly from 0 to 8.4% depending on the geographic regions in which analyses were performed [11, 19, 22–24]. Furthermore, while some studies have suggested that the susceptibility profiles of *C. africana* to antifungal drugs are similar to those of *C. albicans* [25], others have reported different antifungal susceptibility patterns for these species [8, 26]. In light of the above discrepancies concerning *C. africana* prevalence and antifungal susceptibility, the present review and meta-analysis was designed to summarize all of the available data concerning this recent addition to the *C. albicans* species complex.

## Methods

### Search strategy

Two independent researchers conducted bibliographic search in PubMed, Scopus, and Web of Science databases as well as in Google Scholar using keywords or phrases "*Candida africana*", "*C. africana*", "*Candida albicans* complex", "*Candida albicans* sibling species", and "*Candida albicans* cryptic species" and their combinations. Since *Candida africana* was first described as a novel species in 2001 [10], our search covered the literature published in the English language from 2001 to 21$^{st}$ March 2020.

### Study selection

Citations were included into EndNote software version X8, duplicates were deleted and the title and abstract of remaining citations were reviewed to exclude irrelevant articles. For the remaining citations, full texts were downloaded and evaluated. All English language articles with available full texts that reported data on antifungal susceptibility patterns of *Candida*

*africana* and/or prevalence of *Candida africana* within the *Candida albicans* species complex using molecular methods met the inclusion criteria. Conference abstracts, review articles, and articles reporting data other than the susceptibility pattern and/or prevalence of *Candida africana* were excluded. The quality of the selected studies was checked using the STROBE checklist [27]. References cited in the eligible articles were also screened to guarantee the inclusion of all relevant studies.

### Data extraction

Data including the name of the first author, publication year, country, number of *Candida albicans* complex isolates, number of identified *Candida africana* isolates, the source of *Candida africana* isolates, and the minimum inhibitory concentration (MIC) values of various drugs against *Candida africana* isolates were extracted into a pre-prepared excel file by two independent researchers. Corresponding authors of studies reporting only the summary data of antifungal susceptibility pattern such as MIC range, geometric mean (GM), and $MIC_{50}$ were contacted via email for the raw data. In the case of no response, the summarized data of antifungal susceptibility patterns were excluded from the final analysis.

### Data analysis

The pooled estimated prevalence of *C. africana* within the *C. albicans* complex was calculated using Stata software version 14. Variances and their confidence intervals were calculated using exact method. The pooled estimate was between 0 to 1. For studies reporting a prevalence of 0%, Freeman-Tukey double arcsine transformation was used to stabilize variances. Heterogeneity was determined using the $I^2$ statistic which was calculated using the DerSimonian-Laird method. For quantification of heterogeneity, Cochrane Q test was used. In the presence of heterogeneity, random effect model provides better estimates [28, 29], accordingly, we used this model in calculations when heterogeneity was proved to exist. Subgroup analysis was done to define the prevalence of *C. africana* within the *C. albicans* complex in different countries and continents. The presence of publication bias was checked by using the funnel plot and the Begg's test. In the case of asymmetric funnel plot, Trim and Fill method was used to define the number of missing studies and the imputed estimated prevalence. To check for changes in prevalence over time, meta regression was conducted where the year of publication was set as the independent variable. In all calculations p-values <0.05 were considered to be significant.

## Results

A summary of the results of the search strategy is depicted in Fig 1. The original bibliographic search identified 363 articles. An additional 3 articles were identified though examination of all of the literature cited in the retained articles (other sources, Fig 1). After de-duplication and exclusion of irrelevant citations based on the title and abstract, 73 articles were retained for full text evaluation. At this stage, 32 articles were excluded on the basis of the criteria listed in Fig 1 and 41 articles were eligible to be included in the present study (Table 1). Due to the presence of heterogeneity ($I^2$ = 66.02%, 95% CI 44–77, *p*<001), random-effect model was used. The pooled prevalence of *C. africana* within the *C. albicans* complex was 1.67% (95% CI 0.98–2.49) (Fig 2). Data on prevalence were available for 11 countries from 4 continents. Iran (3.02%, 95%CI 1.51–4.92) and Honduras (3.03%, 95% CI 0.83–10.39) had the highest values and Malaysia (0%) had the lowest reported prevalence. (Table 2, S1 Fig and S2 Fig).

As shown in Fig 3, the funnel plot was broadly symmetrical, suggesting the absence of publication bias. This finding was confirmed using Begg's test (Z = 1.26, *p* = 0.215). In meta-

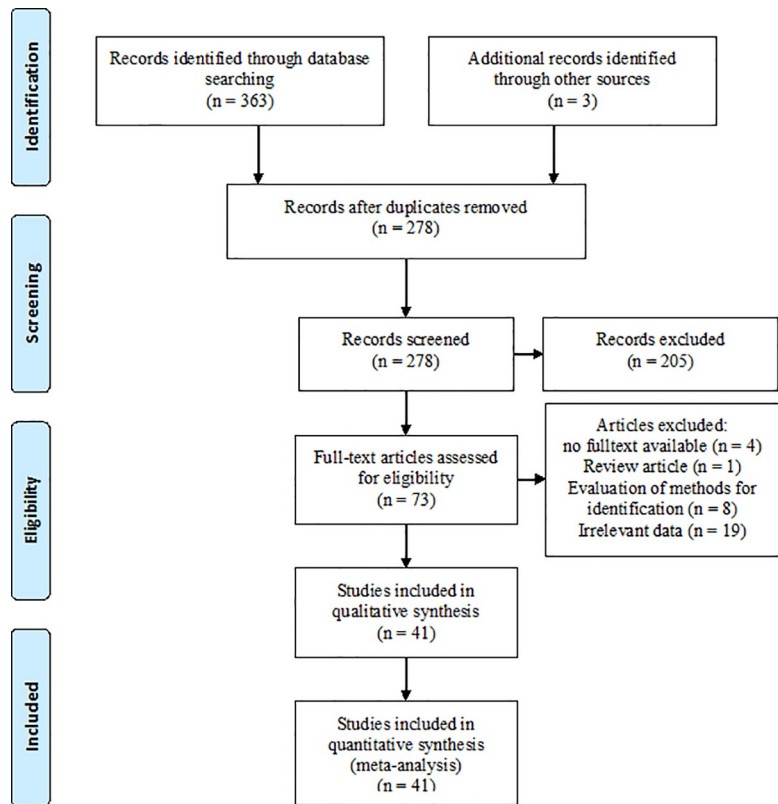

**Fig 1. The PRISMA flow diagram for selection of studies reporting data on intra-*Candida albicans* complex prevalence and/or antifungal susceptibility patterns of *Candida africana* from 2001 to March 2020.**

regression analysis, no evidence for significant change in the prevalence of *C. africana* over time was found (Coefficient = -0.0013, SE = 0.0052, *p* = 0.802) (S3 Fig).

Information on isolation source was available for a total of 167 isolates. Although the vast majority of isolates were from the female genital tract (vagina; n = 155, 92.81%), there were also isolates from patients with balanitis (n = 5, 2.99%), oral lesions (n = 4, 2.39%), and isolates from respiratory, urine, and skin samples (1 isolate each, 0.6%).

MIC values were available for *C. africana* isolates and 16 antifungal drugs including azoles, echinocandins, polyenes, allylamine, and 5-flucytosine. As shown in Table 3, the MIC ranges, $MIC_{50}$ and $MIC_{90}$ and geometric mean values were generally low.

## Discussion

*C. africana*, a member of *C. albicans* species complex, is genetically and phenotypically closely related to *C. albicans*. The pathogenicity of *C. africana* and its impact on the health of humans is poorly understood. Moreover, the global prevalence and antifungal susceptibility profiles of this species are not clearly defined [16, 18, 30]. In this study we tried to provide an overview of the available data published to date on both of these aspects of *C. africana* epidemiology/ biology.

*C. africana* appears to be globally distributed, with an intra-*C. albicans* complex prevalence that varies in different regions and countries [16, 31, 43]. To date, data concerning prevalence are available for 11 countries from 4 continents (Africa, America, Asia, and Europe), with a pooled intra-complex prevalence of 1.67% (95% CI 0.98–2.49). Based on the available

**Table 1. Characteristics of 41 studies reporting data on intra-*Candida albicans* complex prevalence and/or antifungal susceptibility pattern of *Candida africana* which were eligible to be included in the current systematic review and meta-analysis.**

| Reference | Year | Country | No. of *C. albicans* complex/*C. africana* | Source of isolates (N) | Data of antifungal drugs |
|---|---|---|---|---|---|
| Alonso-Vargas et al. [30] | 2008 | Spain | NA/1 | Vagina (1) | Flu, ITR, VRC, KTC, AmB, FLC |
| Borman et al. [17] | 2013 | United Kingdom | 826/15 | Vagina (15) | Flu, ITR, MCN, KTC, CLT, ECN, AmB, NYS |
| Dieng et al. [31] | 2012 | Senegal | 112/3 | Vagina (3) | NA |
| Fakhim et al. [32] | 2020 | Iran | 114/3 | Vagina (3) | Flu, ITR, VRC, AmB, FLC, CSP, ANF, MCF |
| Farahyar et al. [33] | 2020 | Iran | 100/3 | Vagina (3) | Flu, ITR |
| Feng et al. [22] | 2015 | China | 49/0 | - | NA |
| Fontecha et al. [6] | 2019 | Honduras | 66/2 | Vagina (1), Urine (1) | NA |
| Gil-Alonso et al. [8] | 2015 | Spain | NA/2 | Vagina (1), Reference strain (1) | MCF |
| Gil -Alonso et al. [34] | 2015 | Spain | NA/2 | Vagina (1), Reference strain (1) | CSP, ANF, MCF |
| Gil-Alonso et al. [26] | 2016 | Spain | NA/2 | Vagina (1), Reference strain (1) | CSP |
| Gil-Alonso et al. [35] | 2019 | Spain | NA/2 | Vagina (1), Reference strain (1) | ANF |
| Gumral et al. [23] | 2011 | Turkey | 195/0 | - | NA |
| Guzel et al. [36] | 2013 | Turkey | 58/0 | | NA |
| Hashemi et al. [37] | 2019 | Iran | 44/2 | Vagina (2) | NA |
| Hazirolana et al. [25] | 2017 | Turkey | 376/3 | Vagina (3) | Flu, VRC, KTC, AmB, ANF, MCF |
| Hu et al. [38] | 2015 | China | 129/5 | Balanitis (5) | Flu, ITR, VRC, PSC, AmB, FLC, CSP, MCF |
| Kardos et al. [39] | 2017 | Hungary | NA/2 | Vagina (1), Reference strain (1) | MCF |
| Khedri et al. [13] | 2018 | Iran | 74/4 | Oral lesions (4) | Flu, ITR, VRC, AmB, CSP |
| Kova´cs et al. [39] | 2017 | Hungary | NA/2 | Vagina (1), Reference strain (1) | MCF |
| Lortholary et al. [40] | 2007 | France | NA/3 | NA | Flu, VRC, PSC |
| Majdabadi et al. [41] | 2018 | Iran | 40/2 | Vagina (2) | Flu, ITR, AmB |
| Mucci et al. [5] | 2017 | Argentina | 57/0 | | NA |
| Naeimi et al. [19] | 2018 | Iran | 119/10 | Vagina (10) | Flu |
| Ngouana et al. [20] | 2014 | Cameroon | 115/2 | Vagina (2) | Flu, ITR, KTC, AmB |
| Ngouana et al. [42] | 2019 | Cameroon | 115/2 | Vagina (2) | NA |
| Nnadi et al. [43] | 2012 | Italy | 84/2 | Vagina (2) | Flu, VRC, PSC, AmB, CSP, KTC, ITR, FLC |
| Pakshir et al. [44] | 2017 | Iran | 110/0 | - | NA |
| Rezazadeh et al. [45] | 2016 | Iran | 67/4 | Vagina (4) | NA |
| Rezazadeh et al. [45] | 2016 | Iran | NA/4 | NA | Flu, ITR, VRC, PSC, AmB, CSP |
| Romeo et al. [46] | 2009 | Italy | 376/27 | Vagina (27) | NA |
| Romeo et al. [46] | 2009 | Italy | 134/1 | Vagina (1) | NA |
| Scordino et al. [1] | 2019 | Italy | 21/0 | - | NA |
| Shan et al. [3] | 2014 | China | 1014/15 | Vagina (15) | Flu, ITR, NYS, MCN, CLT |
| Sharifynia et al. [14] | 2015 | Iran | 83/1 | Lung (1) | Flu, ITR, AmB, CSP |
| Sharma et al. [16] | 2014 | India | 283/4 | Vagina (4) | Flu, ITR, MCN, VRC, KTC, CLT, PSC, ISC, AmB, FLC, CSP, ANF, MCF, TRB |
| Shokohi et al. [47] | 2018 | Iran | 47/1 | Skin (1) | NA |
| Solimani et al. [24] | 2014 | Iran | 35/0 | - | NA |
| Theill et al. [2] | 2016 | Argentina | 287/1 | Vagina (1) | Flu, ITR, VRC, CLT, AmB, TRB, NYS |

(*Continued*)

**Table 1.** (Continued)

| Reference | Year | Country | No. of *C. albicans* complex/*C. africana* | Source of isolates (N) | Data of antifungal drugs |
|---|---|---|---|---|---|
| Yazdanpanah et al. [11] | 2014 | Malaysia | 98/0 | - | NA |
| Yazdanparast et al. [48] | 2015 | Iran | 114/5 | Vagina (5) | Flu, ITR, VRC, PSC, AmB, CSP, ANF, MCF |
| Zhu et al. [12] | 2019 | China | NA/43 | Vagina (43) | Flu, ITR, MCN, VRC, CLT, BTC, TRC |

Abbreviations: NA: not available, Flu: fluconazole, ITR: itraconazole, VRC: voriconazole, KTC: ketoconazole, AmB: amphotericin B, FLC: 5-fluorocytosine, MCN: miconazole, CLT: clotrimazole, ECN: econazole, NYS: nystatin, CSP: caspofungin, ANF: anidulafungin, MCF: micafungin, PSC: posaconazole, TRB: terbinafine, ISC: isavuconazole, BTC: butoconazole, TRC: terconazole

literature, Malaysia (0.0%; 95% CI 0.0–3.77) has the lowest prevalence. Iran (3.02%; 95% CI 1.51–4.92) and Honduras (3.03%; 95% CI 0.83–10.39) have the highest prevalence. However, since prevalence in Iran was drawn from 12 different studies, it is likely to be a more reliable estimate than the prevalence reported for Honduras, which was based on a single study. Variation in prevalence could be seen between and across continents. For instance, although Iran has the highest prevalence, the prevalence of *C. africana* in a neighboring country, Turkey, is dramatically lower (0.22%; 95% CI 0.0–0.91). It is unclear whether this difference in relative prevalence is the result of an insufficient number of studies in Turkey, genuine local geographical variation, or a combination of both. It is also worth mentioning that the prevalence values reported in the current study are estimated with limited numbers of studies from each country. Data are also lacking for the majority of countries. Thus, the present view might change if there were more studies internationally that addressed the prevalence of *C. africana*.

The intra-complex prevalence of *C. africana* appears to be constant over time. In recent decades, the prevalence of non-*albicans Candida* species has increased [49, 50] and there are reports describing species other than *C. albicans* as being the most common etiologic agents of infection locally [51–53]. However, it seems that a similar scenario has not been occurring within the *C. albicans* species complex since the meta-regression analysis of our data indicates that there is no significant change in the intra-complex prevalence of *C. africana* with the passage of time. However, once again there are caveats to this suggestion. First, it is based on data from a limited number of countries. Moreover, the power of meta-regression analyses is low especially when the number of studies included is low, which is the case in the present study.

Female genital specimens are the most common source of isolation of *C. africana*. Of 167 *C. africana* isolates with available data, the majority (n = 155, 92.81%) were from the vagina. Vulvovaginal candidiasis due to *C. africana* has been reported in various countries [32]. This species was also isolated from cases of balanitis (n = 5, 2.99%) and oral lesions (n = 4, 2.39%), and from respiratory, urine, and skin samples (each 1 isolate, 0.6%), all of which could conceivably become contaminated with vaginal flora or pathogens. The apparent preponderance of *C. africana* for the female genital tract highlights the need for appropriate methods for discrimination of *C. africana* from *C. albicans* complex isolates, especially for vaginal specimens.

Data on antifungal susceptibility of *C. africana* to 16 antifungal drugs are available in the published literature (Table 3). It should be highlighted that the data presented in Table 3 are limited to articles in which detailed results of antifungal susceptibility testing are provided. Other articles that have reported their results as the number of resistant/susceptible isolates or as geometric mean and MIC range (and not the raw MICs) could not be included in Table 3. Similar patterns of susceptibility to various antifungal drugs has been reported for *C. africana* and *C. albicans* [25], while other studies have noted that *C. africana* exhibits a different

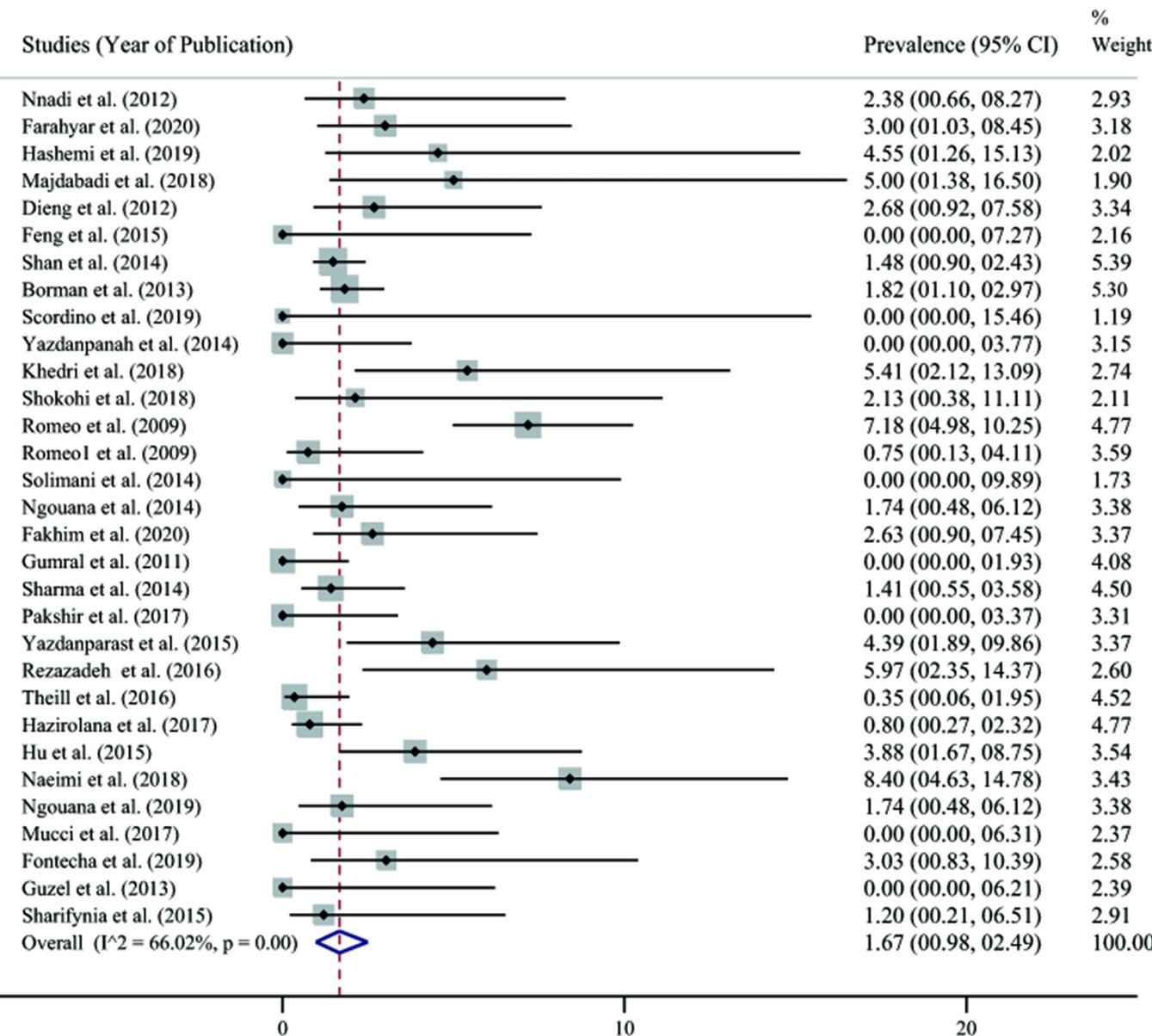

**Fig 2. The forest plot of intra-*Candida albicans* complex prevalence of *Candida africana* based on the reported articles between 2001 to March 2020 (size of squares is representative of the relative weight of studies).**

susceptibility pattern to *C. albicans* [8, 26]. Since there are no specified clinical breakpoints or epidemiological cut-off values (ECVs) for antifungal drugs against *C. africana*, the interpretation of MICs as susceptible/resistant or wild-type/non wild-type is potentially controversial. However, there are reports in which isolates of *C. africana* have been categorized as resistant to itraconazole, 5-flucytosine, terbinafine, fluconazole, and clotrimazole [28, 33, 54]. By applying the clinical breakpoints for *C. albicans* (CLSI M60) [55], the species most closely related to *C. africana*, it could be inferred that almost all isolates of *C. africana* with available MICs for fluconazole, voriconazole, anidulafungin, caspofungin, and micafungin are susceptible to these antifungal drugs. For itraconazole, in contrast to CLSI (M60 supplement) [55] which no longer proposes breakpoints for *Candida* species, the European Committee on Antimicrobial

**Table 2. The pooled intra-*Candida albicans* complex prevalence of *Candida africana* in different countries and continents based on the reported studies between 2001 to March 2020.**

| Continent | Country | Prevalence (%) (95% confidence interval) |
| --- | --- | --- |
| Africa | Cameroon | 1.74 (0.65–4.54) |
| | Senegal | 2.68 (0.87–7.98) |
| | Overall | 2.09 (1–4.32) |
| America | Argentina | 0.11 (0.00–1.04) |
| | Honduras | 3.03 (0.83–10.39) |
| | Overall | 0.51 (0.00–2.46) |
| Asia | China | 1.50 (0.22–3.57) |
| | India | 1.41 (0.55–3.58) |
| | Iran | 3.02 (1.51–4.92) |
| | Malaysia | 0.00 (0.00–3.77) |
| | Turkey | 0.22 (0.00–0.91) |
| | Overall | 1.66 (0.81–2.73) |
| Europe | Italy | 2.33 (0.04–6.79) |
| | United Kingdom | 1.82 (1.10–2.97) |
| | Overall | 2.17 (0.29–5.25) |

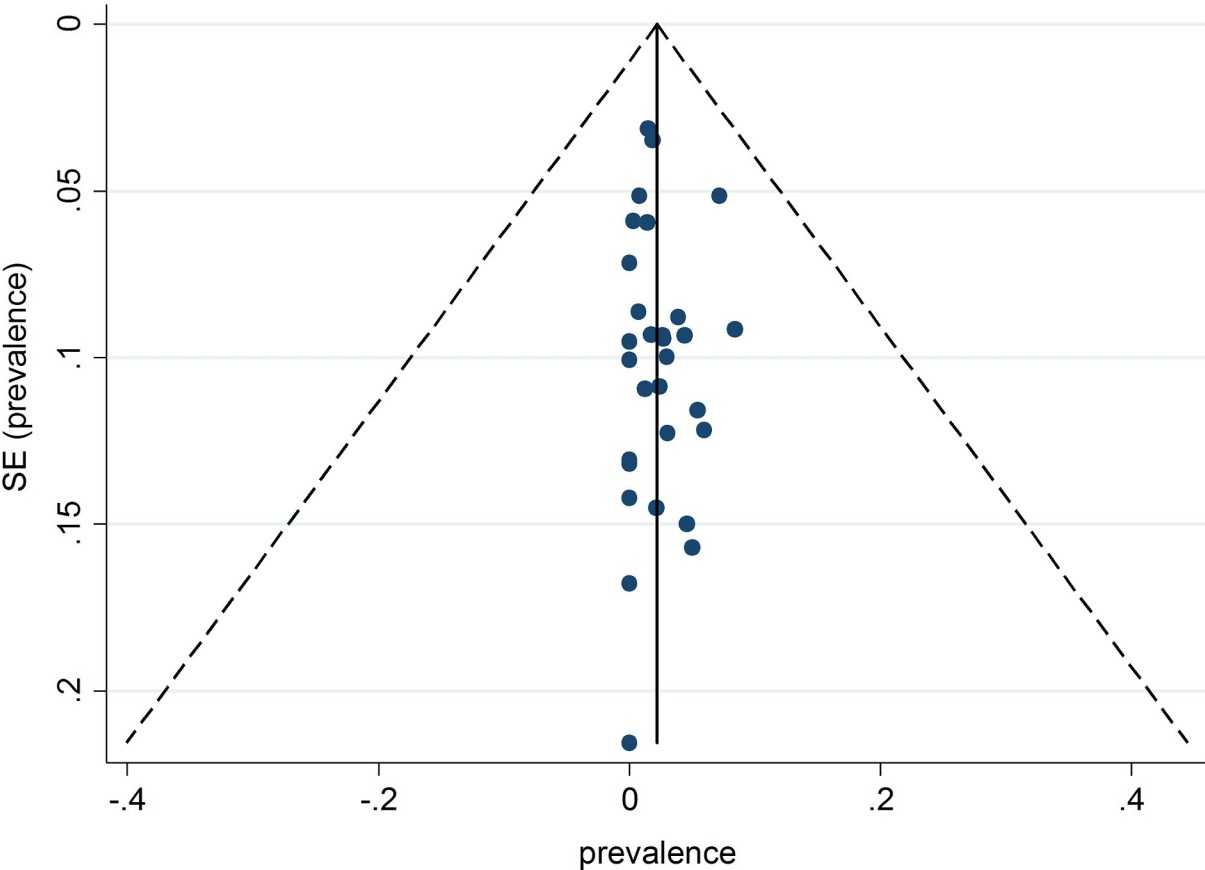

**Fig 3. The funnel plot of available studies reporting data on intra-*Candida albicans* complex prevalence of *Candida africana* between 2001 to March 2020 (each circle is representative of one study).**

**Table 3. The summary of all data reporting antifungal susceptibility patterns of *Candida africana* during 2001 to March 2020 (studies without raw data of minimum inhibitory concentrations are not included).**

| Antifungal drug | No. of isolates with available data | Minimum inhibitory concentration (MIC) values (μg/mL) | | | |
|---|---|---|---|---|---|
| | | MIC range | $MIC_{50}$ | $MIC_{90}$ | Geometric mean |
| Fluconazole | 53 | 0.063–1 | 0.125 | 0.5 | 0.13 |
| Itraconazole | 43 | 0.016–0.25 | 0.031 | 0.125 | 0.031 |
| Voriconazole | 30 | 0.008–0.25 | 0.016 | 0.25 | 0.022 |
| Ketoconazole | 23 | 0.008–2 | 0.063 | 0.063 | 0.04 |
| Posaconazole | 21 | 0.008–0.031 | 0.016 | 0.016 | 0.013 |
| Miconazole | 18 | 0.016–0.063 | 0.063 | 0.063 | 0.046 |
| Clotrimazole | 18 | 0.016–0.25 | 0.063 | 0.063 | 0.048 |
| Econazole | 10 | 0.063–0.063 | 0.063 | 0.063 | 0.063 |
| Isavuconazole | 4 | 0.016–0.016 | 0.016 | 0.016 | - |
| Caspofungin | 27 | 0.008–0.5 | 0.031 | 0.25 | 0.040 |
| Micafungin | 22 | 0.008–0.125 | 0.016 | 0.063 | 0.018 |
| Anidulafungin | 13 | 0.008–0.063 | 0.016 | 0.031 | 0.016 |
| Amphotericin B | 35 | 0.016–8 | 0.125 | 0.5 | 0.113 |
| Nystatin | 15 | 0.031–2 | 1 | 2 | 0.758 |
| 5-flucytosine | 6 | 0.016–0.125 | 0.063 | 0.125 | - |
| Terbinafine | 5 | 1–2 | 2 | 2 | - |

Susceptibility Testing recently published new breakpoints for itraconazole against *C. albicans* and *C. dubliniensis* [29]. Using those breakpoints (>0.06 μg/mL = resistance) 12 out of 43 (27.91%) *C. africana* isolates with available data would be itraconazole-resistant. Further studies will be required to generate MIC data for sufficient numbers of isolates of *C. africana* to allow the establishment of robust species-specific ECVs and clinical breakpoints for this species.

## Conclusion

*C. africana* is a minor species within the *C. albicans* complex with a pooled prevalence of 1.67%. Reports of this species are available from a limited number of countries and further investigations are required internationally to fully address its global distribution. The vagina is the most common human source of *C. africana* and based on clinical breakpoints established for the related *C. albicans*, this species can be inferred to be generally susceptible to most currently available antifungal drugs.

## Supporting information

**S1 Fig. The forest plot of intra-*Candida albicans* complex prevalence of *Candida africana* in different countries based on the reported articles between 2001 to 2020 (size of squares is representative of the relative weight of studies).**
(TIF)

**S2 Fig. The forest plot of intra-*Candida albicans* complex prevalence of *Candida africana* in different continents based on the reported articles between 2001 to 2020 (size of squares is representative of the relative weight of studies).**
(TIF)

**S3 Fig. The meta-regression of intra-*Candida albicans* complex prevalence of *Candida africana* with the time (size of circles is representative of the relative weight of studies;**

**studies with a prevalence of 0% are not shown).**
(TIF)

**S1 Appendix. The search strategy used in PubMed database to find relevant literature.**
(CSV)

**S2 Appendix. The completed PRISMA checklist for systematic reviews.**
(DOC)

## Author Contributions

**Conceptualization:** Shahram Mahmoudi.

**Data curation:** Sanaz Aghaei Gharehbolagh, Bahareh Fallah, Alireza Izadi, Zeinab Sadeghi Ardestani, Pooneh Malekifar, Andrew M. Borman, Shahram Mahmoudi.

**Formal analysis:** Pooneh Malekifar, Shahram Mahmoudi.

**Investigation:** Sanaz Aghaei Gharehbolagh, Bahareh Fallah, Alireza Izadi, Zeinab Sadeghi Ardestani.

**Methodology:** Sanaz Aghaei Gharehbolagh, Shahram Mahmoudi.

**Project administration:** Shahram Mahmoudi.

**Software:** Pooneh Malekifar.

**Supervision:** Shahram Mahmoudi.

**Writing – original draft:** Sanaz Aghaei Gharehbolagh, Bahareh Fallah, Alireza Izadi, Shahram Mahmoudi.

**Writing – review & editing:** Andrew M. Borman, Shahram Mahmoudi.

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
