## [Decision Letter · Decision Letter 0]

16 Jun 2020

PONE-D-20-13736

Distribution, antifungal susceptibility pattern and intra-Candida albicans species complex prevalence of Candida africana: A systematic review and meta-analysis

PLOS ONE

Dear Dr. Mahmoudi,

Thank you for submitting your manuscript to PLOS ONE. After careful consideration, we feel that it has merit but does not fully meet PLOS ONE’s publication criteria as it currently stands. Therefore, we invite you to submit a revised version of the manuscript that addresses the points raised during the review process.

The manuscript by three reviewers including a statistical expert and the reviewers and myself agree that the manuscript provides new information about antifungal susceptibilities of  the less studied pathogen *Candida africana*.  However, prior to a final review the manuscript needs to be proofread for English, preferably by a native English speaker.

The reviewers offer several ways in which this manuscript can be improved and please consider all the points of reviewers 1 and 2.

Reviewer 1 discusses the problems associated with comparing antifungal susceptibilities between species using the current guidelines. Please address this as well as the points referring to labeling of figures. Additionally, Reviewer 1 supplies examples of how the English needs to be modified.

Reviewr 2 has guiding suggestions to clarify metanalysis protocol.

We look forward to receiving your revised manuscript.

Kind regards,

Joy Sturtevant

Academic Editor

PLOS ONE

Journal Requirements:

Additional Editor Comments (if provided):

Reviewers' comments:

Reviewer's Responses to Questions

**Comments to the Author**

1. Is the manuscript technically sound, and do the data support the conclusions?

Reviewer #1: Yes

Reviewer #2: Partly

Reviewer #3: Partly

2. Has the statistical analysis been performed appropriately and rigorously? 

Reviewer #1: Yes

Reviewer #2: Yes

Reviewer #3: Yes

3. Have the authors made all data underlying the findings in their manuscript fully available?

Reviewer #1: Yes

Reviewer #2: Yes

Reviewer #3: No

4. Is the manuscript presented in an intelligible fashion and written in standard English?

Reviewer #1: No

Reviewer #2: No

Reviewer #3: No

5. Review Comments to the Author

Reviewer #1: This article provides a metadata analysis of the less known fungal pathogen C. africana. The paper suffers from a few issues that can be addressed as follows:

The paper needs to be reviewed by a native English speaker as their are many grammatical and wording mistakes. An example of these area as follows:

Sentence structures and English errors:

Line 34-35: sentence should be revised.

Line 62-63: sentence needs to be corrected and susceptibility "to drugs" should be added.

Line 134: "values" not vales.

Line 150: "is" should be added between "It" and "worthy".

Line 155: "this estimated values" should be "this estimated value".

Line 35: Change “which suggesting” to “which suggest”.

Lines 35 – 37: This sentence should be reformulated since it is not clear: “C. africana is a minor species within the C. albicans complex with geographical variation in prevalence and high susceptibility to antifungal drugs”.

Line 44: Change “for infection in human” to “for human infections”.

Line 61: Change “vary” to “varies”.

Line 78: Change “pattern” to “patterns”.

Line 102: Change “remained” to “were retained”.

Line 103: Change “by reasons (fig. 1)” to “according to the reasons mentioned in fig. 1”.

Line 152: Change “if there be more studies” to “if there were more studies”.

Line 157: Change “over the time” to “over time”.

Line 159: Change “agent” to “agents”.

Line 168: Change “such as cases of” to “such as in cases of”.

Line 187: Change “from limited number” to “from a limited number

As far as content please find the below comments that need to be addressed. Especially the issue of antifungal MICs and breakpoints, and which guidelines to use:

1. Version of "Comprehensive Meta-Analysis software" used should be indicated along with the version of any other program or software used.

2. Not all abbreviations are included in the "abbreviation" section under Table 1 such as AmB.

3. Funnel Plot shows only 28 circles while it should include 41 circles resembling all included studies. Please explain this discrepancy.

4. What are the "other sources" mentioned in Figure 1?

5. It would be better to mention some molecular methods in line 59.

6. Titles of all tables and figures should be modified to include legends or symbols, and all needed info so that one can understand them without referring to the text.

7. In the discussion, it is stated that C. africana is susceptible to most antifungal drugs "fluconazole, itraconazole, voriconazole, anidulafungin, caspofungin, micafungin and flucytosine". Other studies show that it is sensitive to conventional antifungals but rather resistant to voriconazole, flucytosine, and terbinafine (Naeimi et al., Journal of Medical Microbiology 2018): This should be addressed in the discussion as it is contradicting info.

The discussion could be slightly improved by not comparing the MICs of itraconazole and flucytosine in Candida africana to those of Candida albicans according to the CLSI M60 guidelines because in these guidelines there is no proof of correct MIC breakpoints for these antifungals with Candida albicans. Moreover, it is suggested in the CLSI M60 guidelines to not take into consideration MIC breakpoints for these antifungals in Candida albicans from previous guidelines where minimal clinical data were used to establish them. For itraconazole, the analysis could be made by using as reference the EUCAST 2018 guidelines.

Do not compare the Candida albicans and Candida africana MICs for flucytosine since no breakpoints for this antifungal exist in both the CLSI M60 and EUCAST 2018 guidelines.

Reviewer #2: I will focus on methods and reporting

Major

1) reading the abstract it was not clear to me what the antifungal aim was - the prevalence aspect was clear

2) similarly at the end of the introduction there is a mention to the "susceptibility pattern" of the species. this is not clear enough the aims needs to be made much more clear and specific at the end of that section.

3) I don't see how publication bias tests are relevant when prevalence is investigated. there is no effect size and as far as I know there is no equivalent test. can the authors expand on this?

4) Meta-analyses of proportions are a bit more complicated since transformations are needed to account for the 0 and 100% limits. Step 1: transformation; step 2: meta-analysis method using standard approach (i.e. inverse variance DerSimonian-Laird); step 3: back-transformation to percentages and plotting. One approach is logit transformation, which is explained in a different context here: http://www.bmj.com/content/352/bmj.i1114. However, a double arcsine transformation is the norm (http://jech.bmj.com/content/early/2013/08/20/jech-2013-203104). Is that what the software you use does? either way, can you clarify please?

5) Meta-regression is a stab in the dark usually and is underpowered to detect anything but massive associations (effectively a regression with X observations, where X is the number of available studies). You should discuss this as a major limitation. Even with 60 or 80 studies, it can provide little insight.

6) Avoid fixed effect models since they under-perform in the presence of ANY heterogeneity. Random-effects (RE) models are more conservative and provide better estimates with wider confidence intervals: http://www.ncbi.nlm.nih.gov/pubmed/11252006 and http://www.ncbi.nlm.nih.gov/pubmed/21148194 . What does the arbitrary 50% cut-off point add? Why is 49% fine and 51% an issue? My point is that a RE model will work better, in the presence of 5% heterogeneity, compared to a FE model!

7) Report the confidence intervals for I^2 (calculated using heterogi or metaan in Stata) as argued in http://www.ncbi.nlm.nih.gov/pubmed/17974687. A simple formula exists in the seminal 2002 Higgins paper that proposed I^2.

Minor

1) why the 2001 start for the literature search?

2) some language corrections are needed.

3) there is a mention to "intra-complex" prevalence in the abstract. I don't know what this is and uncless the authors are very confident about the term and it's just not my area of expertise (which is meta-analysis methods) I'd ask them to revise it.

4) clarify that in the regression the independent variable is year.

5) How was the random-effect model implemented, i.e. how was heterogeneity estimated? There are numerous ways to do so. Did they use the standard DerSimonian-Laird method? If so, please state so. Also there are better performing methods, for example please see https://www.ncbi.nlm.nih.gov/pubmed/28815652 (or http://www.ncbi.nlm.nih.gov/pubmed/23922860) and the metaan command in Stata where these are implemented (https://www.stata-journal.com/article.html?article=st0201).

6) Cochran Q (i.e. chi-square) is notoriously underpowered to detect heterogeneity, especially for small meta-analyses http://www.ncbi.nlm.nih.gov/pubmed/9595615. I would not use

7) the quality of the forest plot is poor. for example there is no negative prevalence, edit the scale.

8) High heterogeneity estimates according to some researchers mean studies should not be meta-analysed. However, I disagree with that assessment for numerous methodological reasons. First heterogeneity is not study size independent: smaller meta-analyses are more often homogeneous and larger studies are heterogeneous (for both methodological and practical reasons). So according to this mantra we will be filtering out the most valuable analyses, the ones that involve many studies, over the much smaller ones. In addition, random effects models can model that heterogeneity and account for it. Finally, large heterogeneity is the norm and it's great if it has been picked up and can be incorporated in the model. It is much more problematic when the underlying heterogeneity is not picked up and studies are "safely" combined under a homogeneity assumption. In other words, small meta-analyses of “homogeneous” studies are much more problematic than large “heterogeneous” ones as evidenced in this http://www.ncbi.nlm.nih.gov/pubmed/23922860. In smaller meta-analyses existing heterogeneity is just not picked up well enough. The uncertainty of the estimate becomes obvious if the CIs for I^2 are reported, as argued in http://www.ncbi.nlm.nih.gov/pubmed/17974687.

Reviewer #3: The manuscript [PONE-D-20-13736] entitled “Distribution, antifungal susceptibility pattern and intra-Candida albicans species complex prevalence of Candida africana: A systematic review and meta-analysis” reviewed and discussed a general view about the prevalence and susceptibility pattern of Candida albicans species complex.

The title, aim and body of manuscript are not concise and coherent. Moreover, there were a surprisingly high number of spelling mistakes considering this standard of writing. Manuscript needs to be a major revision of English editing, preferably by a native speaker/editor.

Authors mentioned

Due to the limited knowledge concerning the prevalence and antifungal susceptibility pattern of this species, a comprehensive study is needed in this regard.

Major comments, manuscript is a superficial paper and has poorly prepared “head and tail”. Based on ref. 30, Candida africana Vulvovaginitis: Prevalence and Geographical Distribution. J Mycol Med. 2020:100966, recently has been already published in this particular region. What is new in this particular paper?

As presented, it does not merit acceptance. It is more suitable for a local journal. It is agreed that this could be a useful contribution to the literature but not just yet-there simply is not enough data to be of any practical use.

Comments to Author:

Interpretation and conclusions: derived from the data.

References: up to date and relevant.

Abstract reflects the content of the paper

Limited number of terms (i.e., key words) used to conduct searches was presented, what may lead to an underestimate of the general picture of the evidence base available for analysis. The number of terms used to conduct searches, including MeSH terms and free terms should be developed.

6. PLOS authors have the option to publish the peer review history of their article (what does this mean?). If published, this will include your full peer review and any attached files.

Reviewer #1: No

Reviewer #2: No

Reviewer #3: No

---

## [Author Response · Author response to Decision Letter 0]

9 Jul 2020

PONE-D-20-13736

Reviewers' comments:

Comments to the Author

1. Is the manuscript technically sound, and do the data support the conclusions?

Reviewer #1: Yes

Reviewer #2: Partly

Reviewer #3: Partly

 2. Has the statistical analysis been performed appropriately and rigorously?

Reviewer #1: Yes

Reviewer #2: Yes

Reviewer #3: Yes

3. Have the authors made all data underlying the findings in their manuscript fully available?

Reviewer #1: Yes

Reviewer #2: Yes

Reviewer #3: No

Response: We are somewhat confused by this comment raised by reviewer #3. All of the data used in this study are presented in full in the accompanying text, tables, figures and supplementary files. The statistical reviewer and reviewer #1 also confirm that all the data are available. 

If reviewer #3 means that our study did not capture all the relevant articles and data because our search strategy is restricted, our response to this concern is given in the section dedicated to the specific issues raised by reviewer #3. 

 4. Is the manuscript presented in an intelligible fashion and written in standard English?

Reviewer #1: No

Reviewer #2: No

Reviewer #3: No

Response: The manuscript has now been fully revised by a native English speaker to improve its quality. 

5. Review Comments to the Author

Reviewer #1: 

This article provides a metadata analysis of the less known fungal pathogen C. africana. The paper suffers from a few issues that can be addressed as follows:

The paper needs to be reviewed by a native English speaker as there are many grammatical and wording mistakes. 

Response: Thank you for your detailed review and for highlighting errors in writing. Based on your comment, the manuscript was referred to native English speaker who is active in the field of Candida research. He has extensively revised the text throughout, and is now included as a co-author. 

An example of these area as follows:

Sentence structures and English errors:

Line 34-35: sentence should be revised.

Response: This sentence has been modified in the revised version.

Line 62-63: sentence needs to be corrected and susceptibility "to drugs" should be added.

Response: Thank you for your suggestion. This was revised in line with your comments.

Line 134: "values" not vales.

Line 150: "is" should be added between "It" and "worthy".

Line 155: "this estimated values" should be "this estimated value".

Line 35: Change “which suggesting” to “which suggest”.

Lines 35 – 37: This sentence should be reformulated since it is not clear: “C. africana is a minor species within the C. albicans complex with geographical variation in prevalence and high susceptibility to antifungal drugs”.

Response: Thank you for pointing out this issue. This was revised to improve its clarity. 

Line 44: Change “for infection in human” to “for human infections”.

Line 61: Change “vary” to “varies”.

Line 78: Change “pattern” to “patterns”.

Line 102: Change “remained” to “were retained”.

Line 103: Change “by reasons (fig. 1)” to “according to the reasons mentioned in fig. 1”.

Line 152: Change “if there be more studies” to “if there were more studies”.

Line 157: Change “over the time” to “over time”.

Line 159: Change “agent” to “agents”.

Line 168: Change “such as cases of” to “such as in cases of”.

Line 187: Change “from limited number” to “from a limited number

Response: Thank you for your suggestions. All the above errors have been corrected in the revised version of the manuscript, and additional textual changes have been introduced to improve readability. 

As far as content please find the below comments that need to be addressed. Especially the issue of antifungal MICs and breakpoints, and which guidelines to use:

1. Version of "Comprehensive Meta-Analysis software" used should be indicated along with the version of any other program or software used.

Response: Thank you for your suggestion. Based on the comments of reviewer #2, we re-analyzed all of the data using Stata software. Versions of both Stata and EndNote programs were added.

2. Not all abbreviations are included in the "abbreviation" section under Table 1 such as AmB.

Response: We apologize for this oversight in the original version. All the abbreviations used in Table 1 are now included in the “abbreviation” section under Table 1. 

3. Funnel Plot shows only 28 circles while it should include 41 circles resembling all included studies. Please explain this discrepancy.

Response: As indicated in Table 1, only a proportion of the 41 articles present data of prevalence. Some articles only included data on antifungal susceptibility testing, and other studies were performed on reference strains. Those studies could not be included in the funnel plot which presents data for prevalence. Because of this, the number of articles in funnel plot is less than the total number of included studies. Please note also that the number of studies presented in funnel plot is actually 31, with several circles almost exactly overlaying other data points.

4. What are the "other sources" mentioned in Figure 1?

Response: The database search returned 363 records. After removing duplicate and irrelevant articles, the full texts of the remaining articles were selected for data extraction. The complete reference lists of all retained articles were also checked to guarantee that we had not missed any articles. Using this procedure we identified three additional records. This has now been clarified at the beginning of the results section.

5. It would be better to mention some molecular methods in line 59.

Response: Thank you for your suggestion. The relevant section has been amended.

6. Titles of all tables and figures should be modified to include legends or symbols, and all needed info so that one can understand them without referring to the text.

Response: Table titles and figure legends have been modified and more information provided so that they are understandable independently of the main text.

7. In the discussion, it is stated that C. africana is susceptible to most antifungal drugs "fluconazole, itraconazole, voriconazole, anidulafungin, caspofungin, micafungin and flucytosine". Other studies show that it is sensitive to conventional antifungals but rather resistant to voriconazole, flucytosine, and terbinafine (Naeimi et al., Journal of Medical Microbiology 2018): This should be addressed in the discussion as it is contradicting info.

Response: Thank you for your comment. The relevant section of the discussion has been revised and specific mention of this conflicting report has been made. 

8. The discussion could be slightly improved by not comparing the MICs of itraconazole and flucytosine in Candida africana to those of Candida albicans according to the CLSI M60 guidelines because in these guidelines there is no proof of correct MIC breakpoints for these antifungals with Candida albicans. Moreover, it is suggested in the CLSI M60 guidelines to not take into consideration MIC breakpoints for these antifungals in Candida albicans from previous guidelines where minimal clinical data were used to establish them. For itraconazole, the analysis could be made by using as reference the EUCAST 2018 guidelines. Do not compare the Candida albicans and Candida africana MICs for flucytosine since no breakpoints for this antifungal exist in both the CLSI M60 and EUCAST 2018 guidelines.

Response: We appreciate this suggestion for improving our work. This part of the discussion has been extensively modified in the revised version and your comments have been addressed. 

Reviewer #2: 

I will focus on methods and reporting

Comment to reviewer: Dear reviewer, we really appreciate your detailed review and we would like to thank you for all of your comments which have helped us to significantly improve the present work. Based on your comments and suggestions, and because the previous software we have used had limitations that meant that we were not able to edit the plots or export them in better formats, we have re-analyzed the entire dataset using Stata software. Because the estimated prevalence values obtained by Stata were slightly different from those produced using the previous software, we checked a third software package which confirmed the results obtained with Stata and indicated that our previous results were slightly over-estimated. All of the prevalence values in the revised manuscript have been corrected, and are now based on the results obtained using Stata. 

Major

1) reading the abstract it was not clear to me what the antifungal aim was - the prevalence aspect was clear

Response: We apologize that this aspect of the original manuscript was not clear. We have now clarified this issue in both the abstract and the introduction. Effectively, a limited number of previous publications had offered conflicting messages regarding the antifungal susceptibility profiles of C. africana as a species, with several reports suggesting that it is likely to be susceptible to most currently employed antifungal drugs, but a least one report suggesting elevated MICs (and possible resistance) to some drugs. Thus, a secondary aim of the current study was to compile all of the available data regarding the susceptibility patterns of this fungus (Table 3) in order to provide a composite picture regarding the minimum inhibitory concentrations of antifungal drugs against this species. We hope that such data will be of use to the intended audience of this article (mycologists and microbiologists).

2) similarly at the end of the introduction there is a mention to the "susceptibility pattern" of the species. this is not clear enough the aims needs to be made much more clear and specific at the end of that section.

Response: Once again, we thank the reviewer for highlighting the lack of clarity in the original manuscript. The offending section has been substantially revised (see also response to point 1 above). 

3) I don't see how publication bias tests are relevant when prevalence is investigated. there is no effect size and as far as I know there is no equivalent test. can the authors expand on this?

Response: Thank you for raising this potential concern. Based on consultations with experts in epidemiology and biostatistics, including in prevalence studies, we believe that publication bias might exist. For example, if the expected prevalence of a variable in a population is 20% but the result obtained in a given study is much different (for instance 2%), this study has a lower chance of being accepted for publication. We believe that this bias is likely to be more extreme regarding the topic of our article. Any study that reported a prevalence of 0% for Candida africana (ie failed to detect any occurences), has a very low chance of being accepted for publication, especially in recent years, as it is likely to be judged as “adding nothing useful to the literature”. Conversely, studies that report C. africana isolates in a particular setting, especially if the reports are accompanied by additional analyses such as antifungal susceptibility testing, genotyping, determining enzymatic activities on different substances etc. will be deemed to generate new and interesting data in addition to a prevalence. Accordingly, in the latter situation, there is a significantly higher chance of this type of study being accepted for publication. 

4) Meta-analyses of proportions are a bit more complicated since transformations are needed to account for the 0 and 100% limits. Step 1: transformation; step 2: meta-analysis method using standard approach (i.e. inverse variance DerSimonian-Laird); step 3: back-transformation to percentages and plotting. One approach is logit transformation, which is explained in a different context here: http://www.bmj.com/content/352/bmj.i1114. However, a double arcsine transformation is the norm (http://jech.bmj.com/content/early/2013/08/20/jech-2013-203104). Is that what the software you use does? either way, can you clarify please?

Response: As discussed above, the software we used in the previous version of this article, although user-friendly, had several limitations. Based on your other comments, the meta-analysis was repeated using metaprop package in Stata software. Variances and their confidence intervals were calculated using exact method. The pooled estimate was from 0 to 1. For studies reporting a prevalence of 0%, Freeman-Tukey Double Arcsine Transformation was used to stabilize the variance.

These data have now been added to the manuscript, and the methods section has been amended to incorporate these changes in methodologies. 

5) Meta-regression is a stab in the dark usually and is underpowered to detect anything but massive associations (effectively a regression with X observations, where X is the number of available studies). You should discuss this as a major limitation. Even with 60 or 80 studies, it can provide little insight.

Response: This concern has now been discussed as a potential significant limitation in the Discussion section of the revised manuscript. 

6) Avoid fixed effect models since they under-perform in the presence of ANY heterogeneity. Random-effects (RE) models are more conservative and provide better estimates with wider confidence intervals: http://www.ncbi.nlm.nih.gov/pubmed/11252006 and http://www.ncbi.nlm.nih.gov/pubmed/21148194 . What does the arbitrary 50% cut-off point add? Why is 49% fine and 51% an issue? My point is that a RE model will work better, in the presence of 5% heterogeneity, compared to a FE model!

Response: Thank you for your invaluable comment. We re-analyzed the data using random-effect in Stata software. Furthermore, using the references you mentioned, we have clarified that in the presence of heterogeneity, random effect was used because in this situation it provides better estimates.

7) Report the confidence intervals for I^2 (calculated using heterogi or metaan in Stata) as argued in http://www.ncbi.nlm.nih.gov/pubmed/17974687. A simple formula exists in the seminal 2002 Higgins paper that proposed I^2.

Reference: Thank you for your comment. The confidence interval for I^2 was calculated and has been added to the manuscript.

Minor

1) why the 2001 start for the literature search?

Response: The literature search was started from 2001 because Candida africana, the species we have focused on, was officially described in 2001 (Mycoses. 2001;44(11-12):437-445).This point has now been made in the revised manuscript. 

2) some language corrections are needed.

Response: The manuscript was revised by a native English speaker. We hope that the English language usage is now much improved. 

3) there is a mention to "intra-complex" prevalence in the abstract. I don't know what this is and uncless the authors are very confident about the term and it's just not my area of expertise (which is meta-analysis methods) I'd ask them to revise it.

Response: Thank you for your comment. Candida africana, the organism which we have focused on, is a member of a complex consisting of several closely-related species, coined the Candida albicans species complex. Thus, when we report a prevalence for Candida africana within this complex, it is the “intra-Candida albicans complex prevalence of Candida africana” ie. we are measuring the “prevalence/proportion of Candida africana within the Candida albicans species complex”. Based on your comment, all “intra-complex(s)” comments used in manuscript have been changed to read “intra-Candida albicans species complex” for clarity and consistency. We did also consult with an expert in epidemiology who advised that it is scientifically correct to use “intra-Candida albicans complex prevalence of Candida africana” in this context. 

4) clarify that in the regression the independent variable is year.

Response: Thank you for your comment. This has been clarified in the revised manuscript. 

5) How was the random-effect model implemented, i.e. how was heterogeneity estimated? There are numerous ways to do so. Did they use the standard DerSimonian-Laird method? If so, please state so. Also there are better performing methods, for example please see https://www.ncbi.nlm.nih.gov/pubmed/28815652 (or http://www.ncbi.nlm.nih.gov/pubmed/23922860) and the metaan command in Stata where these are implemented (https://www.stata-journal.com/article.html?article=st0201).

Response: The heterogeneity was determined using I^2 statistic. Based on your comment (Major 6, above) and our response, in the presence of heterogeneity, random-effect model was used. I^2 was calculated using the standard DerSimonian-Laird method. These issues are clarified in the revised manuscript. 

6) Cochran Q (i.e. chi-square) is notoriously underpowered to detect heterogeneity, especially for small meta-analyses http://www.ncbi.nlm.nih.gov/pubmed/9595615. I would not use

Response: We agree with your comment. As you mentioned, Cochran Q (i.e. chi-square) has a low power for estimation of heterogeneity. However, it should be highlighted that we used this method for quantification of heterogeneity. This has been clarified in the section “Methods”. 

7) the quality of the forest plot is poor. for example there is no negative prevalence, edit the scale.

Response: We agree with your comment. This was another issue that stemmed from the previous software we used in the original version of the manuscript. In response to your previous comments we have re-analyzed all of the data using Stata. As a result, we realized that the original prevalence values calculated using the previous software are indeed over-estimated. Accordingly, all the results are now revised. To ensure the accuracy of the new results, we also analyzed the data using a third software (medcalc) and the results confirmed those obtained with Stata. The manuscript has been modified throughout to capture these changes.

8) High heterogeneity estimates according to some researchers mean studies should not be meta-analysed. However, I disagree with that assessment for numerous methodological reasons. First heterogeneity is not study size independent: smaller meta-analyses are more often homogeneous and larger studies are heterogeneous (for both methodological and practical reasons). So according to this mantra we will be filtering out the most valuable analyses, the ones that involve many studies, over the much smaller ones. In addition, random effects models can model that heterogeneity and account for it. Finally, large heterogeneity is the norm and it's great if it has been picked up and can be incorporated in the model. It is much more problematic when the underlying heterogeneity is not picked up and studies are "safely" combined under a homogeneity assumption. In other words, small meta-analyses of “homogeneous” studies are much more problematic than large “heterogeneous” ones as evidenced in this http://www.ncbi.nlm.nih.gov/pubmed/23922860. In smaller meta-analyses existing heterogeneity is just not picked up well enough. The uncertainty of the estimate becomes obvious if the CIs for I^2 are reported, as argued in http://www.ncbi.nlm.nih.gov/pubmed/17974687.

Response: Thank you for your expert comment. We used random-effect mode in meta-analysis and the confidence interval for I^2 is now provided in the manuscript.

Reviewer #3: 

The manuscript [PONE-D-20-13736] entitled “Distribution, antifungal susceptibility pattern and intra-Candida albicans species complex prevalence of Candida africana: A systematic review and meta-analysis” reviewed and discussed a general view about the prevalence and susceptibility pattern of Candida albicans species complex.

1- The title, aim and body of manuscript are not concise and coherent. Moreover, there were a surprisingly high number of spelling mistakes considering this standard of writing. Manuscript needs to be a major revision of English editing, preferably by a native speaker/editor.

Response: Thank you for your comment. The manuscript has been extensively revised based on your comments and those of reviewers #1 and #2. Furthermore, to improve English language usage, the manuscript was revised by a native English speaker who is active in the field.

2- Authors mentioned Due to the limited knowledge concerning the prevalence and antifungal susceptibility pattern of this species, a comprehensive study is needed in this regard.

Major comments, manuscript is a superficial paper and has poorly prepared “head and tail”. Based on ref. 30, Candida africana Vulvovaginitis: Prevalence and Geographical Distribution. J Mycol Med. 2020:100966, recently has been already published in this particular region. What is new in this particular paper?

Response: We believe that the current study is significantly different to the in press article cited by the reviewer (Fakhim et al., 2020), and is such is a useful addition to the existing literature. The article by Fakhim et al. is a cross-sectional study, despite its title which suggests a review article. In that article, there are no data pertaining to the search strategies employed or search terms, and importantly any inclusion/exclusion criteria. Importantly, their literature search spanned 1991 to 2019, commencing a full decade before Candida africana had been officially described in 2001 (Mycoses. 2001;44(11-12):437-445). Thus, they included data for isolates which had not been robustly identified and were considered to be atypical Candida albicans isolates. The current study explicitly excluded all such isolates from the analyses, ensuring that only data from genuine C. africana isolates was included. 

Additionally, in the present work although the period of search is 10 years less than the article quoted by the reviewer, it identified 41 relevant articles (10 records more than in the cited article) of which 31 have data on prevalence (fig 1). The article referred to by the reviewer was cited in our study, and references cited in it cross-checked to further ensure no relevant article has been missed. 

Finally, in addition to all the above points, our study answers the question “How common Candida africana is within the Candida albicans species complex?”. We provide clear data concerning overall and geographical location-specific data pertaining to intra-complex prevalence of Candida africana whereas the article quoted by the reviewer focused principally on incidence (and not prevalence). 

3- As presented, it does not merit acceptance. It is more suitable for a local journal. It is agreed that this could be a useful contribution to the literature but not just yet-there simply is not enough data to be of any practical use.

Response: We are somewhat confused by this point, especially in light of our responses to the previous comments from this reviewer. The current study is not a regional analysis on prevalence, rather a detailed meta-analysis of reported occurrence/presence and prevalence of C. africana within the C. albicans species complex globally. As such, we do not really understand why the reviewer thinks that this analysis would be more suited to a “local journal”. Based on the points mentioned in response to the above comment, we believe that the data generated in our article would be of interest to researchers working on Candida around the world, especially since we have collated all of the available data from the time of official description of C. africana (2001) to 2020 without any geographical limitations.

4- Interpretation and conclusions: derived from the data.

Response: Thank you for supporting our conclusion.

5- References: up to date and relevant.

Response: Thank you for your support. 

6- Abstract reflects the content of the paper

Response: Thank you for your support.

7- Limited number of terms (i.e., key words) used to conduct searches was presented, what may lead to an underestimate of the general picture of the evidence base available for analysis. The number of terms used to conduct searches, including MeSH terms and free terms should be developed.

Response: Unfortunately, there is no Mesh term for Candida africana. We optimized the current search strategy after numerous modifications and the best results were obtained using the current combination of search terms. In response to your comment, we have since consulted with an experienced librarian and another expert in the subject of this article and they have confirmed the validity of the search terms we have used. Finally, as a further “control” to ensure that our search algorithms did not miss any literature, we also examined all of the literature cited in all of the articles identified in our searches. These points have now been highlighted in the revised manuscript.

---

## [Decision Letter · Decision Letter 1]

20 Jul 2020

Distribution, antifungal susceptibility pattern and intra-Candida albicans species complex prevalence of Candida africana: A systematic review and meta-analysis

PONE-D-20-13736R1

Dear Dr. Mahmoudi,

We’re pleased to inform you that your manuscript has been judged scientifically suitable for publication and will be formally accepted for publication once it meets all outstanding technical requirements.

Kind regards,

Joy Sturtevant

Academic Editor

PLOS ONE

Additional Editor Comments (optional):

Reviewers' comments:

Reviewer's Responses to Questions

**Comments to the Author**

1. If the authors have adequately addressed your comments raised in a previous round of review and you feel that this manuscript is now acceptable for publication, you may indicate that here to bypass the “Comments to the Author” section, enter your conflict of interest statement in the “Confidential to Editor” section, and submit your "Accept" recommendation.

Reviewer #1: All comments have been addressed

Reviewer #2: All comments have been addressed

Reviewer #3: All comments have been addressed

2. Is the manuscript technically sound, and do the data support the conclusions?

Reviewer #1: Yes

Reviewer #2: Yes

Reviewer #3: Yes

3. Has the statistical analysis been performed appropriately and rigorously? 

Reviewer #1: Yes

Reviewer #2: Yes

Reviewer #3: Yes

4. Have the authors made all data underlying the findings in their manuscript fully available?

Reviewer #1: Yes

Reviewer #2: Yes

Reviewer #3: Yes

5. Is the manuscript presented in an intelligible fashion and written in standard English?

Reviewer #1: Yes

Reviewer #2: Yes

Reviewer #3: Yes

6. Review Comments to the Author

Reviewer #1: All comments were addressed in a satisfactory manner.

Just a few minor grammatical mistakes need to be fixed:

Lines 31 and 127: Change "Honduran" to "Honduras"

Line 214: Change "wile-type" to "wild-type"

Line 215: Change "has been" to "have been"

Reviewer #2: I am happy with the changes to the manuscript and the authors' responses. Well done, I hope you feel the review process improved the quality of the manuscript.

Reviewer #3: (No Response)

7. PLOS authors have the option to publish the peer review history of their article (what does this mean?). If published, this will include your full peer review and any attached files.

Reviewer #1: No

Reviewer #2: No

Reviewer #3: **Yes: **Hamid Badali

---

## [Editor Report · Acceptance letter]

6 Aug 2020

PONE-D-20-13736R1 

Distribution, antifungal susceptibility pattern and intra-Candida albicans species complex prevalence of Candida africana: A systematic review and meta-analysis 

Dear Dr. Mahmoudi:

I'm pleased to inform you that your manuscript has been deemed suitable for publication in PLOS ONE. Congratulations! Your manuscript is now with our production department. 

Kind regards, 

on behalf of

Dr. Joy Sturtevant 

Academic Editor

PLOS ONE